# Comparison of Patients Classified as High-Risk between International Cardiovascular Disease Primary Prevention Guidelines

**DOI:** 10.3390/jcm13154379

**Published:** 2024-07-26

**Authors:** Niamh Chapman, Monique Breslin, Zhen Zhou, James E. Sharman, Mark R. Nelson, Richard J. McManus

**Affiliations:** 1School of Health Science, Faculty of Medicine and Health, University of Sydney, Camperdown, NSW 2050, Australia; 2Menzies Institute for Medical Research, University of Tasmania, Hobart, TAS 7000, Australia; 3School of Public Health and Preventive Medicine, Monash University, Melbourne, VIC 3800, Australia; 4Nuffield Department of Primary Care Health Sciences, University of Oxford, Radcliffe Observatory Quarter, Oxford OX2 6GG, UK

**Keywords:** absolute cardiovascular disease risk, risk prediction, international guideline development

## Abstract

**Background:** Cardiovascular disease (CVD) primary prevention guidelines classify people at high risk and recommended for pharmacological treatment based on clinical criteria and absolute CVD risk estimation. Despite relying on similar evidence, recommendations vary between international guidelines, which may impact who is recommended to receive treatment for CVD prevention. **Objective:** To determine the agreement in treatment recommendations according to guidelines from Australia, England and the United States. **Methods:** Cross-sectional analysis of the National Health and Nutrition Examination Survey (n = 2647). Adults ≥ 40 years were classified as high-risk and recommended for treatment according to Australia, England and United States CVD prevention guidelines. Agreement in high-risk classification and recommendation for treatment was assessed by Kappa statistic. **Results:** Participants were middle aged, 49% were male and 38% were white. The proportion recommended for treatment was highest using the United States guidelines (n = 1318, 49.8%) followed by the English guidelines (n = 1276, 48.2%). In comparison, only 26.6% (n = 705) of participants were classified as recommended for treatment according to the Australian guidelines. There was moderate agreement in the recommendation for treatment between the English and United States guidelines (κ = 0.69 [0.64–0.74]). In comparison, agreement in recommendation for treatment was minimal between the Australian and United States guidelines (κ = 0.47 [0.43–0.52]) and weak between the Australian and English guidelines (κ = 0.50 [0.45–0.55]). **Conclusions:** Despite similar evidence underpinning guidelines, there is little agreement between guidelines regarding the people recommended to receive treatment for CVD prevention. These findings suggest greater consistency in high-risk classification between CVD prevention guidelines may be required.

## 1. Introduction

Cardiovascular disease (CVD) is the leading cause of death and disability worldwide [1]. International clinical guidelines include two key components recommended to identify people for treatment for the primary prevention of CVD. The first component is typically a set of high-risk clinical criteria that result in the recommendation for treatment without the need for a risk assessment. The rationale for these criteria is that current CVD risk algorithms do not appropriately account for the contribution of these criteria to increased risk of future events because the underlying algorithms often exclude such patients. The second component is the assessment of multiple risk factors to identify individuals at high risk for CVD, known as absolute CVD risk [2,3,4,5,6,7,8,9]. In practice, this approach uses multivariable risk prediction tools, which incorporate demographic, clinical and biological factors to estimate the risk of a cardiovascular event over a given time period. The future risk of a cardiovascular event is often referred to as a ‘score’ that is used in combination with a treatment threshold to guide clinical management. National CVD prevention guidelines differ in the prediction tools used, the cardiovascular outcomes used and associated thresholds, as well as the target population recommended for treatment, and high-risk clinical characteristics [10].

Implementing guideline recommendations for the primary prevention of CVD as part of clinical care enables the identification and subsequent management of individuals at risk who otherwise would remain undetected. Guidelines provide recommendations for lifestyle modifications that are applicable at all levels of risk alongside recommendations to identify individuals for treatment to lower the future risk of CVD. These recommendations typically include dietary changes [11], increasing physical activity [12], and smoking cessation [13]. Although superior to single risk factor management strategies, absolute CVD risk prediction models and classification thresholds may result in under- or over-treating individuals [14,15,16]. Those at high risk or moderate risk with additional risk factors or screening are typically recommended for pharmacological treatment alongside lifestyle changes. Given that absolute CVD risk algorithms rely on traditional risk factors, several guidelines include more innovative approaches to identify atherosclerosis and increased risk that may otherwise be missed [8,15]. However, the implementation of existing guideline recommendations for CVD prevention remains poor. The length and complexity of guidelines has been suggested as a major barrier to uptake [17], which may also result in further disparities in those identified as high-risk and recommended for treatment to prevent CVD.

## 2. Objective

The aim of this study was to compare the agreement in individuals classified as high-risk and recommended for pharmacological treatment according to CVD primary prevention guidelines from Australia, England and the United States of America.

## 3. Methods

### 3.1. Study Overview

This cross-sectional hypothetical case study used data from the United States National Health and Nutrition Examination Survey (NHANES) 2011–2012, of which the methods have been reported previously [18]. Adults aged 40 years and older were included if blood pressure readings were available and there was no history of previous myocardial infarction or stroke (as guidelines used in the current study are for primary and not secondary prevention); see Appendix A for participant flow. Self-reported data included age, sex, smoking status, ethnicity, medications and disease status. Height, weight, blood pressure and cholesterol were measured during biomedical examinations using standardised methods [18].

Blood pressure, lipid and/or CVD primary prevention guidelines were used from Australia, England and the United States [2,4,5,6,7,8]. Absolute CVD risk estimation was undertaken among all participants using the risk prediction equation recommended in each guideline. Table 1 provides an overview of the guideline recommendations applied to classify individuals as ‘high risk’, where pharmacological treatment with statin or anti-hypertensive medication is recommended. First, participants were classified according to high-risk clinical characteristics. Second, absolute CVD risk estimation was undertaken with the risk prediction equation and associated threshold recommended in each guideline. Finally, participants classified as high-risk were combined to provide the total recommended for pharmacological treatment.

### 3.2. High-Risk Clinical Characteristics

Participants were classified as high-risk if specific clinical characteristics were present where treatment for CVD prevention is recommended in guidelines without the need for absolute CVD risk estimation. These specific characteristics are selected because absolute CVD risk estimation may under-estimate risk among this population. As an example, a participant with diabetes that is 60 years of age or older would be classified as high-risk according to the Australian guidelines, even if the estimated absolute CVD risk is less than 15% (Table 1).

### 3.3. Estimation of Absolute CVD Risk

Absolute CVD risk scores were estimated using the risk prediction tool and threshold for each country-specific guideline (Table 1). The United States guidelines recommend the pooled cohort equations [8]; Australian guidelines recommend an adjusted 5-year Framingham risk equation [2]; English guidelines recommend QRISK2 [6]. Age, sex, systolic blood pressure, smoking status, diabetes status and cholesterol were common across all risk prediction tools. For dichotomous variables, such as chronic kidney disease, rheumatoid arthritis and family history of CVD, missing data were coded as the absence of disease [19]. In accordance with Australian guidelines, an age of 74 years was used for absolute CVD risk estimation in those aged 74 years or older. The age range shown in Table 1 was used for absolute CVD risk estimation for the English and United States guidelines.

#### 3.3.1. Systolic Blood Pressure

The systolic blood pressure used for the analysis was the average of the second and third of three sequential measurements. If both the second and third blood pressure readings were not available, then any available reading was used in the analyses.

#### 3.3.2. Social Deprivation in QRISK 2

A measure of social deprivation known as the ‘Townsend score’ is used in QRISK2 absolute CVD risk estimation. Townsend scores were created for participants using the required four variables: unemployment, overcrowding as a measure of material living conditions, owner-occupied accommodation, and car ownership as an indicator of income [20]. Car ownership was not available in the NHANES data, so income was used as a dichotomous variable above or below the mean as a proxy. The component scores were standardised using the ᴢ-score technique and the resultant scores were summed to provide a composite score with equal weight given to each variable.

### 3.4. Data Analysis

Four demographic characteristics were presented for the unweighted sample. Agreement was assessed by the head-to-head comparison of each guideline with respect to the proportion of overlap in classification of high risk and recommendation for treatment, and as assessed by Cohen’s Kappa statistic. The level of agreement was defined as κ of 0–0.20 = none, 0.21–0.39 = minimal, 0.40–0.59 = weak, 0.60–0.79 = moderate, 0.80–0.90 = strong and >0.90 = almost perfect [21].

## 4. Results

### 4.1. Participant Characteristics

Participant characteristics are reported in Table 2. On average, participants were middle-aged, normotensive and nearly 17% had diabetes. For medications, 41% of participants were taking anti-hypertensive medication and over a quarter were taking statins.

### 4.2. Participants Classified as High-Risk and Recommended for Treatment According to CVD Prevention Guidelines

Figure 1 shows the proportion of participants classified as high-risk according to clinical characteristics, the absolute CVD risk estimation among those without high-risk clinical characteristics, and the total recommended for treatment. Overall, the United States guidelines classified the greatest proportion of participants as high-risk (n = 1318, 49.8%), followed by the English guidelines (n = 1276, 48.2%). Conversely, only 26.6% (n = 705) of participants were recommended for treatment according to the Australian guidelines. High-risk classification by each clinical characteristic and absolute CVD risk estimation threshold is shown in Appendix A, which highlights that diabetes and blood pressure were the main contributing risk factors to the recommendation for treatment based on high-risk clinical characteristics.

### 4.3. Agreement between CVD Primary Prevention Guidelines

Table 3 reports the agreement in participants classified as high-risk according to clinical characteristics, absolute CVD risk estimation among those without high-risk clinical characteristics, and the total recommended for treatment between two guidelines. There was moderate agreement in those classified as high-risk and recommended for treatment between the English and United States guidelines. Conversely, the level of agreement between the Australian guidelines and the English and United States guidelines was more varied.

As shown in Table 3, there was moderate agreement in high-risk classification according to clinical characteristics between the Australian and English guidelines. However, there was weak agreement between the Australian and United States guidelines, with only 45.6% of participants (n = 357) classified as high-risk in both guidelines.

There was minimal agreement in high-risk classification according to absolute CVD risk estimation between the Australian and English guidelines, but weak agreement between the Australian and United States guidelines. Only a quarter were high risk in both the Australian and English guidelines and more than a third were high risk in both the Australian and United States guidelines when using absolute CVD risk estimation and threshold (Table 3). In comparison, there was moderate agreement between the English and United States guidelines, with nearly two thirds of participants classified as high-risk in both guidelines (Table 3). When applied to the total sample rather than only those that did not have high-risk clinical characteristics, the agreement in high-risk classification according to absolute CVD risk was weak (rather than minimal) between the Australian guidelines and both the English and United States guidelines (κ = 0.42 [0.37–0.47] and κ = 0.53 [0.47–0.59], respectively), but remained moderate between the English and United States guidelines (κ = 0.74 [0.69–0.80]).

## 5. Discussion

The main finding of this study is that the classification of CVD risk and subsequent recommendation for treatment to prevent CVD varies substantially between guidelines in Australia, England and the United States. The findings demonstrate disparate agreement in risk classification according to clinical characteristics and absolute CVD risk estimation. The prevention of CVD is cost-effective, and whilst policy makers may choose to set thresholds based on available resources, these findings highlight disparities in the individuals recommended to receive treatment. These inconsistencies are largely due to definitions of risk and clinical characteristics that cannot be explained by either the underlying evidence or population differences. Overall, these findings may suggest a need for international consensus on the clinical characteristics that indicate high risk without the need for risk estimation and the role of absolute risk thresholds in CVD primary prevention guidelines.

There is a strong rationale for using absolute CVD risk estimation to identify high-risk patients as those with the greatest benefit from preventive treatment [22,23,24]. More than 360 CVD risk prediction equations [14] have been published since the Framingham Heart Study investigators first pioneered this approach in 1991. As previously noted, there is an excess of models to predict CVD risk, but there is a need for the external validation and cross-comparison of existing CVD models to determine generalisability and performance by association with clinical endpoints [15,16]. However, as CVD risk prediction equations are used in conjunction with associated thresholds and high-risk clinical characteristics specified in guidelines, future work comparing risk equations should also consider these criteria. Comparing absolute CVD risk estimation equations without associated thresholds does not adequately consider how such equations are used in practice to identify individuals likely to benefit from treatment to prevent CVD. In this study, we observed poor agreement in recommendations for treatment according to absolute CVD risk estimation. However, absolute CVD risk classification is only one element of CVD primary prevention guidelines. As demonstrated in this study, high-risk clinical characteristics play a key role in identifying individuals at high risk for treatment.

Guidelines for the primary prevention of CVD typically include clinical characteristics deemed to classify an individual as high-risk without the need for absolute CVD risk estimation [2,4,8]. Such characteristics are typically decided by expert consensus formed by interpreting non-definitive evidence [2,6,8]. Therefore, these characteristics may bear little relationship to the absolute CVD risk thresholds used elsewhere in the guidelines. The present study shows that many individuals would be recommended for treatment based solely on these high-risk clinical characteristics, with a lack of consistency in who would be recommended for treatment based on these criteria between guidelines. Of the individuals classified as high-risk based on clinical characteristics from the Australian and United States guidelines, less than half of these individuals were classified as high risk in *both* guidelines. Altogether, this highlights the lack of high-quality data to determine such high-risk clinical criterial for inclusion in guidelines, and the impact on the implementation of guideline-directed care is unknown.

### 5.1. Implications for Clinical Practice

A clear disparity exists between CVD primary prevention guidelines with the potential to result in the inappropriate treatment or a missed opportunity for treatment for CVD prevention. Many national guidelines have adopted an absolute CVD risk estimation approach to guide treatment decisions, and perhaps now is the time to develop an international consensus on high-risk clinical characteristics and align absolute CVD risk thresholds, or at least the manner in which the thresholds are determined. The present study has shown that clinical characteristics account for a large proportion of individuals classified as high-risk, ranging from 17 to 26% across the guidelines, whereas the thresholds for absolute CVD risk estimation recommended more than twice as many individuals for treatment according to the English and United States guidelines compared to the Australian guidelines. While absolute CVD risk thresholds are predominantly driven by CVD prevalence and cost-effectiveness analyses relevant to each country, there is opportunity to develop international consensus on the identification of high-risk patients and the definition of CVD for primary prevention. Indeed, there may be rationale for recommendations to align where international evidence is applicable to national-level health priorities to support a unified approach to CVD prevention.

Although many guidelines now recommend absolute CVD estimation, implementation in practice is limited [25,26], with general practitioners typically focusing on single risk factor management [27]. Treatment based on QRISK2 and associated thresholds has been shown to be superior to treatment according to blood pressure and could prevent a fifth more CVD deaths than the combined blood pressure criteria and absolute CVD risk score [28].

Previous work has shown that general practitioners find guidelines difficult to use [26,29], which may contribute to the poor implementation and uptake of treatment according to guideline recommendations in primary care. There have been calls for countries to align blood pressure and lipid management guidelines into a single, overarching CVD prevention guideline to facilitate uptake among practitioners [30]. As demonstrated in this study, combining two guidelines (blood pressure and/or lipid) to determine recommendations for CVD prevention treatment was required for all three countries. In addition, high-risk clinical characteristics related to blood pressure and diabetes recommended nearly a quarter of participants for treatment according to the United States guidelines, which suggests that single risk factors play a major role within risk-factor-specific guidelines even where absolute CVD risk estimation is recommended. Altogether, there is an opportunity to develop an overarching CVD prevention guideline based on absolute CVD risk to rationalise recommendations and increase clinical utility.

### 5.2. Strengths and Weakness

This study applied a systematic approach combining blood pressure and lipid guidelines between each country to assess the overall impact of cardiovascular prevention strategies using a large, nationally representative sample of participants. A limitation of this study is the theoretical nature, and although others have modelled the impact of hypertension guidance, we cannot draw conclusions from this work on which guideline is superior for CVD prevention [28]. In addition, we did not have access to a detailed clinical history, which limited our ability to apply several guideline recommendations. Given the differences in classification for treatment recommendations observed in this study, further research, with clinical outcomes including CVD events, is warranted to compare the performance of CVD prevention guidelines for optimum risk prediction.

While a systematic approach to comparing guidelines was undertaken, guidelines include many caveats, which could not be accounted for within this analysis. For example, both the Australian and English guidelines classify individuals with moderate to severe chronic kidney disease as high-risk. In the absence of clinical record data, we classified all participants with any level of chronic kidney disease as high-risk. Finally, absolute CVD risk guidelines were recently updated in Australia and England after this analysis was completed [31,32]. However, in this study, guidelines from a similar time period have been compared, and the updated guidelines still include recommendations for treatment based on high-risk clinical characteristics. The latter point is important because these high-risk clinical characteristics were observed to be a major reason for the lack of concordance between the guidelines, suggesting that similar observations may be found in a comparative analysis of the updated guidelines. Future work to determine the agreement in classification of high risk and recommendations for treatment among more contemporary guidelines with outcome data would be worthwhile.

### 5.3. Future Perspectives

Guideline recommendations and thresholds for treatment are regularly updated based on emerging evidence and to adapt contemporary health needs [33]. This study showed that guidelines from the same time period have different recommendations that result in disparities in the individuals classified as high-risk and recommended for treatment for primary prevention of CVD. The clinical implication of this discrepancy is not answered in this present study and is an important consideration for future work. As shown by Damen and colleagues via a systematic review in 2016 [14], many studies have shown that the precision of absolute CVD risk estimation to predict future CVD events is limited. In recent years, there has been growing interest in and evidence for more advanced approaches to risk stratification for the primary prevention of CVD, including the use of biomarkers, imaging techniques and personalised interventions using artificial intelligence. [15,34] However, the implementation of guideline recommendations for CVD prevention has remained a stubborn barrier in primary care [35]. As highlighted by a 2022 AHA statement [36], future work using implementation science methods may help address the evidence-to-practice gap for current recommendations and future innovations.

### 5.4. Conclusions

This study has highlighted inconsistencies in the identification of individuals at high risk of CVD and recommended for treatment between international CVD primary prevention guidelines. This disparity is not limited to the estimation of absolute CVD risk and associated thresholds but extends to the clinical characteristics that indicate high risk without the need for absolute CVD risk. These findings suggest that greater consistency in high-risk classification between CVD prevention guidelines may be required to inform daily clinical practice for CVD prevention.

## Figures and Tables

**Figure 1 jcm-13-04379-f001:**
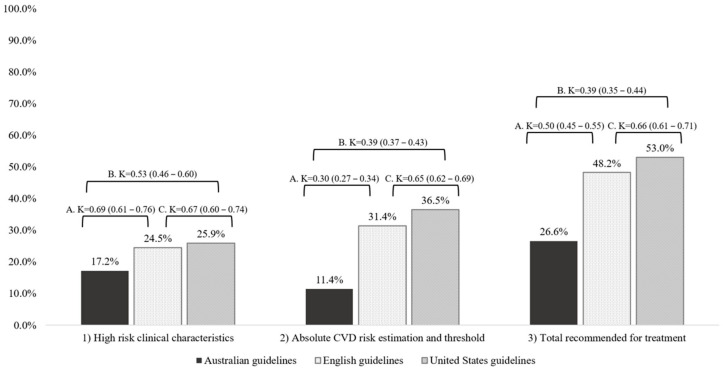
Proportion of individuals classified as high-risk of cardiovascular disease (CVD) according to (1) high-risk clinical characteristics without the need for absolute CVD risk estimation. (2) Absolute CVD risk estimation and associated threshold. (3) Total recommended for treatment for CVD prevention based on CVD primary prevention guidelines from Australia, England and the United States. Data from adults aged 40 years or older from National Health and Nutrition Examination Survey (NHANES) 2011–2012 were used for analyses (n = 2647). Comparison between (A) Australian and English guidelines, (B) Australian and United States guidelines, and (C) English and United States guidelines.

**Table 1 jcm-13-04379-t001:** Overview of the cardiovascular disease primary prevention guidelines in Australia, England and the United States used to classify participants from the United States (US) National Health and Nutrition Examination Survey at high risk of cardiovascular disease.

	Australia	England	United States of America
Guidelines used	Guidelines for the management of absolute cardiovascular risk (2012) [2] Guideline for the diagnosis and management of hypertension in adults (2016) [3]	Cardiovascular disease: risk assessment and reduction including lipid modification (2014) [5,6] Hypertension in adults: diagnosis and management (2019) [4]	Guidelines on the primary prevention of cardiovascular disease (2019) [8] Guidelines for the prevention, detection, evaluation and management of high BP in adults (2017) [7]
Predicted outcomes			
Myocardial infarction	Yes	-	Yes
Coronary heart disease *	Yes	Yes	Yes
Stroke	Yes	Yes	Yes
Transient ischaemic attack	Yes	Yes	-
Angina	Yes	-	-
Heart failure	Yes	-	-
Peripheral vascular disease	Yes	-	-
High-risk clinical characteristics that indicate high risk without the need for absolute cardiovascular risk assessment			
Diabetes	Aged ≥60 years;	Aged <80 years	Aged 40–75 years
BP-specific	SBP ≥ 140 mmHg and diabetes ≥ 180/110 mmHg;	≥150/90 mmHg aged > 80 years; ≥160/100 mmHg aged < 80 years;	BP ≥ 140/90 mmHg and absolute CVD risk < 10%;
Chronic kidney disease	Yes	Yes	BP ≥ 130/80 mmHg
Familial hypercholesterolaemia	Yes	Yes	N/A
Cholesterol	Total > 7.5 mmol/L	-	LDL ≥ 190 mg/dL
Absolute CVD risk estimation			
Age range	45–74 years	40–84 years	40–75 years
Risk model	FRE	QRISK-2	PCE
Risk estimation range	5-year	10-year	10-year
Risk threshold (%)	>15	10 *	7.5 ** 20
BP adjusted risk threshold	≥140/90 mmHg and absolute CVD risk score 10–15%;	BP ≥ 140/90 mmHg and absolute CVD risk ≥ 10%	BP > 130/80 mmHg and absolute CVD risk > 10%
Risk factors used in absolute CVD risk assessment model			
Age	Yes	Yes	Yes
Sex	Yes	Yes	Yes
Cholesterol ^X^	Yes	Yes	Yes
Systolic BP	Yes	Yes	Yes
Smoking status	Yes	Yes	Yes
Diabetes status	Yes	Yes	Yes
Treated BP	-	Yes	-
Body mass index	-	Yes	-
Family history of CVD	-	Yes	-
Rheumatoid arthritis	-	Yes	-
Atrial fibrillation	-	Yes	-
Chronic kidney disease ^‡^	-	Yes	-
Ethnicity	-	Yes	Yes
Socioeconomic status	-	Yes	-

Abbreviations: FRE, Framingham risk equation; PCE, pooled cohort equations; BP, blood pressure. * Refers to coronary heart disease death in Australian and United States guidelines and all coronary heart disease in English guidelines. ** Intermediate risk is recommended for treatment in the presence of risk-enhancing factors that include family history of CVD, LDL cholesterol 160–189 mg/dL, metabolic syndrome, chronic kidney disease, chronic inflammatory conditions including psoriasis and rheumatoid arthritis, and history of premature menopause, gestational diabetes or preeclampsia. ^X^ Total and high-density lipoprotein cholesterol. ^‡^ Moderate or severe chronic kidney disease (persistent proteinuria or estimated glomerular filtration rate < 60 mL/min/1.73 m^2^ (English and United States guidelines) or <45 mL/min/1.73 m^2^ (Australian guidelines).

**Table 2 jcm-13-04379-t002:** Participant characteristics in adults aged 40–84 years from National Health and Nutrition Examination Survey (NHANES) 2011–2012 (n = 2647).

Characteristics	
Age (years)	59 ± 12
Male (%)	1299 (49.1)
Smoking status (% yes)	397 (15.0)
Body mass index (kg/m^2^)	29.3 ± 6.8
Total cholesterol (mmol/L)	5.2 ± 1.1
High density lipoprotein cholesterol (mmol/L)	1.4 ± 0.4
Systolic blood pressure (mmHg)	127.1 ± 18.3
Diastolic blood pressure (mmHg)	71.8 ± 13.5
Blood pressure ≥ 140/90 mmHg	624 (23.6)
Diabetes (%)	446 (16.9)
Medications	
Anti-hypertensive, n (%)	1089 (41.1)
Statin, n (%)	684 (25.8)
Ethnicity	
White, n (%)	1008 (38.1)
Black, n (%)	711 (26.9)
Asian, n (%)	337 (12.7)
Other, n (%)	591 (22.3)
Absolute CVD risk score (% ± SD)	
5-year Framingham risk equation	7.1 ± 6.7
10-year QRISK2	12.2 ± 12.1
10-year Pooled cohort equation	13 ± 15

**Table 3 jcm-13-04379-t003:** Proportion and agreement of participants classified as high-risk and recommended for treatment according to cardiovascular disease primary prevention guidelines from Australia, England and the United States in adults aged over 40 years (n = 2647).

	A. Australian and English Guidelines	B. Australian and United States Guidelines	C. English and United States Guidelines
High-Risk Clinical Characteristics			
Proportion classified as high-risk in both guidelines. n (%)	413 (59.9)	357 (45.6)	501 (60.2)
Agreement in high-risk classification. κ (95% CI)	0.69 (0.61–0.76)	0.53 (0.46–0.60)	0.67 (0.60–0.74)
Agreement level	Moderate	Weak	Moderate
Absolute CVD Risk Estimation and Threshold			
Proportion classified as high-risk in both guidelines. n (%)	175 (24.9)	234 (36.1)	454 (56.3)
Agreement in high-risk classification. κ (95% CI)	0.30 (0.27–0.34)	0.46 (0.39–0.53)	0.63 (0.56–0.70)
Agreement level	Minimal	Weak	Moderate
Total Recommended for Treatment			
Proportion classified as high-risk in both guidelines. n (%)	666 (50.7)	664 (48.9)	1095 (73.1)
Agreement in high-risk classification. κ (95% CI)	0.50 (0.45–0.55)	0.47 (0.43–0.52)	0.69 (0.64–0.74)
Agreement level	Weak	Weak	Moderate

The proportion of people classified as high-risk in both guidelines is among those that were classified as high-risk in either guideline to indicate the concordance in risk classification between guidelines. The level of agreement was defined as κ of 0–0.20 = none, 0.21–0.39 = minimal, 0.40–0.59 = weak, 0.60–0.79 = moderate, 0.80–0.90 = strong and >0.90 = almost perfect [17].

## Data Availability

Publicly available data was used for this study. All data are available for the Nutrition Health and Nutrition Examination Survey (NHANES Questionnaires, Datasets, and Related Documentation (cdc.gov)).

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
