# Peer review of "Comparison of Patients Classified as High-Risk between International Cardiovascular Disease Primary Prevention Guidelines"

_jcm, 2024, doi:10.3390/jcm13154379_

Round 1
Reviewer 1 Report
Comments and Suggestions for Authors
While Chapman, N. et al. try to compare the cardiovascular disease primary prevention guidelines between Australia, England, and the United States, most of the guidelines they reviewed for their studies are outdated; hence, the study will not be beneficial to the readership unless the manuscript is revised significantly with updated guidelines for all the 3 countries being compared.
Example
In Table 1: According to
https://www.mja.com.au/journal/2024/220/9/2023-australian-guideline-assessing-and-managing-cardiovascular-disease-risk#5
https://www.mja.com.au/journal/2023/219/4/evidence-supporting-choice-new-cardiovascular-risk-equation-australia
Most of the Risk factors used in absolute CVD risk assessment model for Australia have changed, and some of the factors indicated “-” in the manuscript are now ‘Yes’.
It is recommended that the investigator revise the manuscript according to the Latest Guidelines for all 3 countries and cite the appropriate references.
P.S. All the sources below indicated most of the guidelines utilized in the study are outdated
https://www.nice.org.uk/guidance/ng238
https://www.bmj.com/content/381/bmj.p1028
https://www.heartfoundation.org.au/for-professionals/guideline-for-managing-cvd
Author Response
Reviewer 1
Comment 1.1 While Chapman, N. et al. try to compare the cardiovascular disease primary prevention guidelines between Australia, England, and the United States, most of the guidelines they reviewed for their studies are outdated; hence, the study will not be beneficial to the readership unless the manuscript is revised significantly with updated guidelines for all the 3 countries being compared.
Example
In Table 1: According to
https://www.mja.com.au/journal/2024/220/9/2023-australian-guideline-assessing-and-managing-cardiovascular-disease-risk#5
https://www.mja.com.au/journal/2023/219/4/evidence-supporting-choice-new-cardiovascular-risk-equation-australia
Most of the Risk factors used in absolute CVD risk assessment model for Australia have changed, and some of the factors indicated “-” in the manuscript are now ‘Yes’.
It is recommended that the investigator revise the manuscript according to the Latest Guidelines for all 3 countries and cite the appropriate references.
P.S. All the sources below indicated most of the guidelines utilized in the study are outdated
https://www.nice.org.uk/guidance/ng238
https://www.bmj.com/content/381/bmj.p1028
https://www.heartfoundation.org.au/for-professionals/guideline-for-managing-cvd
Response 1.1 Thank you for reviewing our manuscript. We acknowledge that the guidelines for absolute CVD risk assessment have been updated. We considered updating the analysis, however we have found this to not be possible because the new algorithm from Australia (adapted from PREDICT) has not yet been made publicly available. Moreover, in Australia and the United States, the hypertension guidelines that have been referred to are both the most up to date documents available for those countries.
We have updated the manuscript to acknowledge this limitation as shown below.
Page 16:
While a systematic approach to comparing guidelines was undertaken, guidelines include many caveats, which could not be accounted for within this analysis. For example, both the Australian and English guidelines classify individuals with moderate to severe chronic kidney disease as high risk. In the absence of clinical record data, we classified all participants with any level of chronic kidney disease as high risk. Finally, absolute CVD risk guidelines have been recently updated in Australia and England after this analysis was completed.32,33 However, in this study guidelines from a similar time period have been compared and the updated guidelines still include recommendations for treatment based on high risk clinical characteristics. The latter point is important because these high risk clinical characteristics were observed to be a major reason for lack of concordance between the guidelines, suggesting that similar observations may be found in a comparative analysis of the updated guidelines. Future work to determine the agreement in classification of high risk and recommendation for treatment among more contemporary guidelines with outcome data would be worthwhile.
Reviewer 2 Report
Comments and Suggestions for Authors
The authors present a study comparing cardiovascular disease prevention guidelines between Australia, England, and the United States, aiming to compare agreements in classifying high-risk patients and patients recommended for pharmacological therapy. The study is well-designed and conducted, and the results are helpful for health providers.
Suggestions:
For the sake of preciseness, I would extend the title (with a limiter), making it such as: “Comparison of cardiovascular disease primary prevention guidelines between Australia, England and the United States: Focus on the high risk patients identification and the need for medical treatment”
In describing the methods, I would use a “hypothetical case study” rather than a theoretical study.
In the discussion section, it would be worth mentioning that actual recommendations and thresholds are moving targets based on everchanging evidences. Also, since all mentioned guidelines (including the European Society of Cardiology guidelines that were not used in this article) are being updated in regular (3-4 year intervals), it would be interesting to see whether there is a trend to converging (national and international) recommendations over time – what need is suggested by this article.
Author Response
Reviewer 2
Comment 1. The authors present a study comparing cardiovascular disease prevention guidelines between Australia, England, and the United States, aiming to compare agreements in classifying high-risk patients and patients recommended for pharmacological therapy. The study is well-designed and conducted, and the results are helpful for health providers.
Response 1. Thank you for undertaking this review and for your feedback.
Comment 2. For the sake of preciseness, I would extend the title (with a limiter), making it such as: “Comparison of cardiovascular disease primary prevention guidelines between Australia, England and the United States: Focus on the high risk patients identification and the need for medical treatment”
Response 2. The title has been updated as suggested and as shown below.
Comparison of cardiovascular disease primary prevention guidelines between Australia, England and the United States: Focus on the high risk patients identification and the need for medical treatment
Comment 3. In describing the methods, I would use a “hypothetical case study” rather than a theoretical study.
Response 3. The methods have been updated as suggested. As follows:
Page 4: This cross-sectional theoretical hypothetical case study used data from the United States National Health and Nutrition Examination Survey (NHANES) 2011-2012, of which the methods have been reported previously.13
Comment 4. In the discussion section, it would be worth mentioning that actual recommendations and thresholds are moving targets based on everchanging evidences. Also, since all mentioned guidelines (including the European Society of Cardiology guidelines that were not used in this article) are being updated in regular (3-4 year intervals), it would be interesting to see whether there is a trend to converging (national and international) recommendations over time – what need is suggested by this article.
Response 4. As suggested, the discussion has been updated to describe how guidelines are subject to change over time based on emerging evidence, as shown below. We agree that it would be interesting to see if recommendations become more consistent over time.
Page 14: While absolute CVD risk thresholds are predominantly driven by CVD prevalence and cost-effectiveness analyses relevant to each country, there is opportunity to develop international consensus on the identification of high risk patients, and the definition of CVD for primary prevention. Indeed, there may be rationale for recommendations to align where international evidence is applicable to national level health priorities to support a unified approach to CVD prevention.
And
Page 15: Guideline recommendations and thresholds for treatment are regularly updated based on emerging evidence and to adapt contemporary health needs.34 This study showed that guidelines from the same time period have different recommendations that results in disparities in the individuals classified as high risk and recommended for treatment for primary prevention of CVD. The clinical implication of this discrepancy is not answered in this present study and is an important consideration for future work. As shown by Damen and colleagues via systematic review in 2016,14 many studies have shown that the precision of absolute CVD risk estimation to predict future CVD events is limited.
Reviewer 3 Report
Comments and Suggestions for Authors
Dear Authors, I have read your manuscript “Comparison of cardiovascular disease primary prevention guidelines between Australia, England and the United States” with interest.
In my opinion, these are the adjustments which should be made to increase the value of your manuscript:
1. In the Introduction section, it is recommended to add more detailed information about the subject discussed in the study - cardiovascular disease primary prevention - general information, components and its importance in clinical practice. It will be helpful to use the guidelines described in this manuscript as well as this recent published article about primary cardiovascular prevention doi.org/10.1016/j.atherosclerosis.2024.117579. After this, please, describe in more detail the objectives of the study.
2. Please, clarify why the latest European Guideline - 2021 ESC Guidelines on cardiovascular disease prevention in clinical practice: Developed by the Task Force for cardiovascular disease prevention in clinical practice with representatives of the European Society of Cardiology and 12 medical societies With the special contribution of the European Association of Preventive Cardiology (EAPC) https://doi.org/10.1093/eurheartj/ehab484 - was not included in the study. This Guideline would complement the study and give it a holistic meaning.
3. In the manuscript, the authors used a very old bibliography, it is recommended to review and add recent studies and articles.
4. Please, add future perspectives.
5. In the conclusions section, indicate the practical significance of your study for further clinical daily practice.
Comments on the Quality of English LanguageMinor editing of English language required.
Author Response
Reviewer 3
Comment 3.1. The authors present a study comparing cardiovascular disease prevention guidelines between Australia, England, and the United States, aiming to compare agreements in classifying high-risk patients and patients recommended for pharmacological therapy. The study is well-designed and conducted, and the results are helpful for health providers.
Comment 3.1. Thank you for providing your feedback.
Comment 3.2. In the Introduction section, it is recommended to add more detailed information about the subject discussed in the study - cardiovascular disease primary prevention - general information, components and its importance in clinical practice. It will be helpful to use the guidelines described in this manuscript as well as this recent published article about primary cardiovascular prevention doi.org/10.1016/j.atherosclerosis.2024.117579. After this, please, describe in more detail the objectives of the study.
Response 3.2. As suggested, we have updated the introduction of the manuscript as shown below:
Page 3: Cardiovascular disease (CVD) is the leading cause of death and disability worldwide.1 International clinical guidelines include two key components recommended to identify people for treatment for the primary prevention of CVD. The first component is typically high risk clinical criteria that result in the recommendation for treatment without the need for risk assessment. The rationale for these criteria is that current CVD risk algorithms do not appropriately account for the contribution of these criteria to increased risk of future events. The second component is the assessment of multiple risk factors to identify individuals at high risk for CVD, known as absolute CVD risk.2–9 In practice this approach uses multivariable risk prediction tools which incorporate demographic, clinical and biological factors to estimate the risk of a cardiovascular event over a given time period. The future risk of a cardiovascular event is often referred to as a ‘score’ that is used in combination with a treatment threshold to guide clinical management. National CVD prevention guidelines differ in the prediction tools used and associated thresholds, as well as the target population recommended for treatment, and high risk clinical characteristics.10
Implementing guideline recommendations for primary prevention of CVD as part of clinical care enables the identification and subsequent management of individuals at risk who otherwise would remain undetected. Guidelines provide recommendations for lifestyle modifications that are applicable at all levels of risk alongside recommendations to identify individuals for treatment to lower future risk of CVD. These recommendations typically include dietary changes,11 increasing physical activity,12 and smoking cessation.13 Although superior to single risk factor management strategies, absolute CVD risk prediction models and classification thresholds may result in under- or over-treating individuals.14–16 Those at high risk or moderate risk with additional risk factors or screening are typically recommended for pharmacological treatment alongside lifestyle changes. Given that absolute CVD risk algorithms rely on traditional risk factors, several guidelines include more innovative approaches to identify atherosclerosis and increased risk that may otherwise be missed.8,15 However, implementation of existing guideline recommendations for CVD prevention remains poor. Length and complexity of guidelines has been suggested as a major barrier to uptake,17 which may also result in disparities in those identified at high risk and recommended for treatment to prevent CVD.
OBJECTIVE
The aim of this study was to compare the agreement in individuals classified as high risk and recommended for pharmacological treatment according to CVD primary prevention guidelines from Australia, England and the United States of America.
Comment 3.3. Please, clarify why the latest European Guideline - 2021 ESC Guidelines on cardiovascular disease prevention in clinical practice: Developed by the Task Force for cardiovascular disease prevention in clinical practice with representatives of the European Society of Cardiology and 12 medical societies With the special contribution of the European Association of Preventive Cardiology (EAPC) https://doi.org/10.1093/eurheartj/ehab484 - was not included in the study. This Guideline would complement the study and give it a holistic meaning.
Response 3.3. We fully agree that the 2021 ESC Guideline is an important document for the prevention of cardiovascular disease, especially for the European context. The original concept of this paper was to also use national health survey data from Australia, England and the United States and test each guideline with each dataset. However, there were significant issues with harmonising and comparing data across these samples. For this reason, we maintained the focus on those three countries and the respective guidelines but undertook the analysis in one sample. It would be worthwhile to compare the ESC guideline to other national guidelines within European to determine the agreement in risk classification and recommendation for treatment. However, that was beyond the scope of this present study.
Comment 3.4. In the manuscript, the authors used a very old bibliography, it is recommended to review and add recent studies and articles.
Response 3.4. The bibliography has been updated throughout with contemporary articles. All new additions are shown in red. Of the 36 references in the updated bibliography, 15 were published since 2020, 16 were published between 2015 and 2020, and 5 were published earlier than 2015 with the earliest published in 2008. However, all publications included that were published prior to 2015 were needed for the methods and need to be reported for the work to be reproducible.
The updated bibliography is pasted at the bottom of this response document for ease of review.
Comment 3.5. Please, add future perspectives.
Response 3.5. As suggested, we have added a future perspective section to the discussion as follows:
Pages 15-16:
FUTURE PERSPECTIVES
Guideline recommendations and thresholds for treatment are regularly updated based on emerging evidence and to adapt contemporary health needs.34 This study showed that guidelines from the same time period have different recommendations that results in disparities in the individuals classified as high risk and recommended for treatment for primary prevention of CVD. The clinical implication of this discrepancy is not answered in this present study and is an important consideration for future work. As shown by Damen and colleagues via systematic review in 2016,14 many studies have shown that the precision of absolute CVD risk estimation to predict future CVD events is limited. In recent years there is growing interest in and evidence for more advanced approaches to risk stratification for primary prevention of CVD including the use of biomarkers, imaging techniques and personalised interventions using artificial intelligence.15,35 However, implementation of guideline recommendations for CVD prevention has remained a stubborn barrier in primary care.36 As highlighted by a 2022 AHA statement,37 future work using implementation science methods may help address the evidence-to-practice gap for current recommendations and future innovations.
Comment 3.6. In the conclusions section, indicate the practical significance of your study for further clinical daily practice.
Response 3.6. As suggested, we have added a statement about the relevance of the study findings for daily clinical practice and highlighted that further work is needed as shown below.
Pages 16-17:
CONCLUSION
This study has highlighted inconsistencies in the identification of individuals at high risk of CVD and recommended for treatment between international CVD primary prevention guidelines. This disparity is not limited to the estimation of absolute CVD risk and associated thresholds, but extends to the clinical characteristics that indicate high risk without the need for absolute CVD risk. These findings suggest greater consistency in high risk classification between CVD prevention guidelines may be required to inform daily clinical practice for CVD prevention.
Updated bibliography:
References:
- Vos, T. et al. Global burden of 369 diseases and injuries in 204 countries and territories, 1990–2019: a systematic analysis for the Global Burden of Disease Study 2019. The Lancet 396, 1204–1222 (2020).
- National Vascular Disease Prevention Alliance. Guidelines for the Management of Absolute Cardiovascular Disease Risk. (2012).
- Gabb, G. M. et al. Guideline for the diagnosis and management of hypertension in adults — 2016. Med. J. Aust. 205, 85–89 (2016).
- National Institute for Health and Care Excellence, (NICE). Hypertension in adults: diagnosis and management. https://www.nice.org.uk/guidance/ng136 (2019).
- National Institute of Clinical Excellence. Cardiovascular Disease: Risk Assessment and Reduction, Including Lipid Modification. Clinical Guideline [CG181]. https://www.nice.org.uk/guidance/cg181/chapter/1-recommendations#identifying-and-assessing-cardiovascular-disease-cvd-risk-2 (2014).
- National Institute for Health and Care Excellence. Cardiovascular disease prevention overview - NICE Pathways. https://pathways.nice.org.uk/pathways/cardiovascular-disease-prevention#path=view%3A/pathways/cardiovascular-disease-prevention/cardiovascular-disease-prevention-overview.xml&content=view-index (2019).
- Whelton, P. K. et al. 2017 ACC/AHA/AAPA/ABC/ACPM/AGS/APhA/ASH/ASPC/NMA/PCNA Guideline for the Prevention, Detection, Evaluation, and Management of High Blood Pressure in Adults: A Report of the American College of Cardiology/American Heart Association Task Force on Clinical Practice Guidelines. J. Am. Coll. Cardiol. 71, e127–e248 (2018).
- Arnett, D. K. et al. 2019 ACC/AHA Guideline on the Primary Prevention of Cardiovascular Disease: A Report of the American College of Cardiology/American Heart Association Task Force on Clinical Practice Guidelines. Circulation (2019) doi:10.1161/CIR.0000000000000678.
- Visseren, F. L. J. et al. 2021 ESC Guidelines on cardiovascular disease prevention in clinical practice: Developed by the Task Force for cardiovascular disease prevention in clinical practice with representatives of the European Society of Cardiology and 12 medical societies With the special contribution of the European Association of Preventive Cardiology (EAPC). Eur. Heart J. 42, 3227–3337 (2021).
- Stewart, J., Manmathan, G. & Wilkinson, P. Primary prevention of cardiovascular disease: A review of contemporary guidance and literature. JRSM Cardiovasc. Dis. 6, (2017).
- Guasch-Ferré, M. & Willett, W. C. The Mediterranean diet and health: a comprehensive overview. J. Intern. Med. 290, 549–566 (2021).
- Kraus, W. E. et al. Physical Activity, All-Cause and Cardiovascular Mortality, and Cardiovascular Disease. Med. Sci. Sports Exerc. 51, 1270–1281 (2019).
- van Trier, T. J. et al. Lifestyle management to prevent atherosclerotic cardiovascular disease: evidence and challenges. Neth. Heart J. Mon. J. Neth. Soc. Cardiol. Neth. Heart Found. 30, 3–14 (2022).
- Damen, J. A. A. G. et al. Prediction models for cardiovascular disease risk in the general population: systematic review. BMJ i2416 (2016) doi:10.1136/bmj.i2416.
- Barkas, F. et al. Advancements in risk stratification and management strategies in primary cardiovascular prevention. Atherosclerosis 395, 117579 (2024).
- Brown, S. et al. Evidence supporting the choice of a new cardiovascular risk equation for Australia. Med. J. Aust. 219, (2023).
- Kränkel, N. et al. Do we practice what we preach? Implementation of cardiovascular prevention strategies in 13 European countries between 2011 and 2021: a statement of the European Association of Preventive Cardiology of the ESC. Eur. J. Prev. Cardiol. 31, e65–e70 (2024).
- Dohmann, J. C., Burt, V. & Mohadjer, L. National Health and Nutrition Examination Survey: Sample Design, 2011-2014. (2014).
- Hippisley-Cox, J. et al. Predicting cardiovascular risk in England and Wales: prospective derivation and validation of QRISK2. BMJ 336, 1475–1482 (2008).
- Yousaf, S. & Bonsall, A. UK Townsend Deprivation Scores from 2011 Census Data.Pdf. http://statistics.digitalresources.jisc.ac.uk.s3.amazonaws.com (2017).
- McHugh, M. L. Interrater reliability: the kappa statistic. Biochem. Medica 22, 276–282 (2012).
- Silverman, M. G. et al. Association Between Lowering LDL-C and Cardiovascular Risk Reduction Among Different Therapeutic Interventions: A Systematic Review and Meta-analysis. JAMA 316, 1289–1297 (2016).
- Ettehad, D. et al. Blood pressure lowering for prevention of cardiovascular disease and death: a systematic review and meta-analysis. Lancet Lond. Engl. 387, 957–967 (2016).
- Sheppard, J. P. et al. Benefits and Harms of Antihypertensive Treatment in Low-Risk Patients With Mild Hypertension. JAMA Intern. Med. 178, 1626–1634 (2018).
- Bonner, C. et al. General practitioner support needs to implement cardiovascular disease risk assessment and management guidelines: Qualitative interviews. AJGP. (In press). AJGP (2024).
- Bonner, C., Fajardo, M. A., Doust, J., McCaffery, K. & Trevena, L. Implementing cardiovascular disease prevention guidelines to translate evidence-based medicine and shared decision making into general practice: theory-based intervention development, qualitative piloting and quantitative feasibility. Implement. Sci. 14, 86 (2019).
- Chapman, N. et al. General practitioners maintain a focus on blood pressure management rather than absolute cardiovascular disease risk management. J. Eval. Clin. Pract. 27, 1353–1360 (2021).
- Herrett, E. et al. Eligibility and subsequent burden of cardiovascular disease of four strategies for blood pressure-lowering treatment: a retrospective cohort study. The Lancet 394, 663–671 (2019).
- Ju, I. et al. General practitioners’ perspectives on the prevention of cardiovascular disease: systematic review and thematic synthesis of qualitative studies. BMJ Open 8, (2018).
- Karmali, K. N. & Lloyd-Jones, D. M. Global Risk Assessment to Guide Blood Pressure Management in Cardiovascular Disease Prevention. Hypertension 69, e2–e9 (2017).
- Nelson, M. R. et al. 2023 Australian guideline for assessing and managing cardiovascular disease risk. Med. J. Aust. Online first, (2024).
- Samarasekera, E. J., Clark, C. E., Kaur, S., Patel, R. S. & Mills, J. Cardiovascular disease risk assessment and reduction: summary of updated NICE guidance. BMJ 381, p1028 (2023).
- Gabbay, J. & le May, A. Mindlines: making sense of evidence in practice. Br. J. Gen. Pract. 66, 402–403 (2016).
- Chung, R. et al. Using Polygenic Risk Scores for Prioritizing Individuals at Greatest Need of a Cardiovascular Disease Risk Assessment. J. Am. Heart Assoc. 12, e029296 (2023).
- Tuzzio, L. et al. Barriers to implementing cardiovascular risk calculation in primary care: alignment with the Consolidated Framework for Implementation Research. Am. J. Prev. Med. 60, 250–257 (2021).
- Moise, N. et al. Leveraging Implementation Science for Cardiovascular Health Equity: A Scientific Statement From the American Heart Association. Circulation 146, e260–e278 (2022).
Round 2
Reviewer 1 Report
Comments and Suggestions for Authors
N/A
Author Response
Comment 1.1 N/A
Response 1.1 Thank you for undertaking peer review of our manuscript.
Reviewer 2 Report
Comments and Suggestions for Authors
Thank you for accepting suggestions and updating the overall appearance of this interesting article. Well-done manuscript. Congratulations.
Author Response
Comment 1. Thank you for accepting suggestions and updating the overall appearance of this interesting article. Well-done manuscript. Congratulations.
Response 1. Thank you for undertaking this review and for your feedback which enabled us to improve the manuscript.
Reviewer 3 Report
Comments and Suggestions for Authors
I agree with the changes made, which significantly improve the quality of the manuscript.
However, the new manuscript title is too long, it is recommended to short and change the title.
Comments on the Quality of English LanguageMinor editing of English language required.
Author Response
Comment 3.1. I agree with the changes made, which significantly improve the quality of the manuscript. However, the new manuscript title is too long, it is recommended to short and change the title.
Response 3.1. Thank you for providing your feedback. As suggested, we have shortened the title from 27 words to 14 words as shown below:
Comparison of patients classified at high risk between international cardiovascular disease primary prevention guidelines between Australia, England and the United States: Focus on the high risk patients identification and the need for medical treatment